# User-Specified Local Differential Privacy in Unconstrained Adaptive Online Learning

**Dirk van der Hoeven**
Mathematical Institute
Leiden University
Leiden, 2333 CA
dirkvderhoeven@gmail.com

## Abstract

Local differential privacy is a strong notion of privacy in which the provider of the data guarantees privacy by perturbing the data with random noise. In the standard application of local differential privacy the distribution of the noise is constant and known by the learner. In this paper we generalize this approach by allowing the provider of the data to choose the distribution of the noise without disclosing any parameters of the distribution to the learner, under the constraint that the distribution is symmetrical. We consider this problem in the unconstrained Online Convex Optimization setting with noisy feedback. In this setting the learner receives the subgradient of a loss function, perturbed by noise, and aims to achieve sublinear regret with respect to some competitor, without constraints on the norm of the competitor. We derive the first algorithms that have adaptive regret bounds in this setting, i.e. our algorithms adapt to the unknown competitor norm, unknown noise, and unknown sum of the norms of the subgradients, matching state of the art bounds in all cases.

## 1 Introduction

In learning, a natural tension exists between learners and the providers of data. The learner aims to make optimal use of the data, perhaps even at the cost of the privacy of the providers. To nevertheless ensure sufficient privacy the provider can add random noise to the data that he sends to the learner. This idea is called $\epsilon$-local differential privacy (Wasserman and Zhou, 2010; Duchi et al., 2014) and the standard implementation has constant $\epsilon$ for all providers. However, not all providers care equivalently about their privacy (Song et al., 2015). Some providers may wish to aid the learner in making optimal use of their data, while other providers value their privacy over helping the learner. For instance, celebrities might care more for their privacy than others because they want to preserve the privacy they have left. To complicate things further, the providers of the data may not wish to reveal how much they care about their privacy, because when privacy levels differ between providers these privacy levels become privacy sensitive themselves. Furthermore, not all parts of the data are equally privacy sensitive. For example, tweets are already publicly available, but browsing history may contain sensitive information that should be kept private. To capture these varying privacy constraints we allow each provider to choose how much noise is added for each dimension of the data.

In this paper, we consider these problems in the Online Convex Optimization (OCO) setting (Hazan, 2016) with local differential privacy guarantees. The OCO framework is a popular and successful framework to design and analyse many algorithms used to train machine learning models. The OCO setting proceeds in rounds $t = 1, \ldots, T$. In a given round $t$ the learner is to provide a prediction $\boldsymbol{w}_t \in \mathbb{R}^d$. An adversary then chooses a convex loss function $\ell_t$ and sends a subgradient $\boldsymbol{g}_t \in \partial \ell_t(\boldsymbol{w}_t)$ to the learner. We work with an unconstrained domain for $\boldsymbol{w}$, which has recently grown in popularity

(see McMahan and Orabona (2014); Foster et al. (2015); Orabona and Pál (2016); Foster et al. (2017); Cutkosky and Boahen (2017); Kotłowski (2017); Cutkosky and Orabona (2018); Foster et al. (2018); Jun and Orabona (2019)). We aim to develop online learning methods that make the best use of data providers who wish to help the learner while at the same time guaranteeing the desired level of privacy for providers that care about their privacy, without knowing how much each each provider adds to the data.

We consider the local differential privacy model with varying levels of privacy unknown to the learner. Differential privacy (Dwork and Roth, 2014) is a privacy model that is used in many recent machine-learning applications. The local differential privacy model is a variant of differential privacy in which the learner can only access the data of the provider via noisy estimates (Wasserman and Zhou, 2010; Duchi et al., 2014). The local differential privacy model with varying levels of privacy appeared before in Song et al. (2015), but with known levels of noise and only two levels of noise.

Learning in our setting is modelled by the OCO framework with noisy estimates of the subgradient (see also Jun and Orabona (2019)). To ensure local differential privacy the provider adds zero-mean noise $\boldsymbol{\xi}_t \in \mathbb{R}^d$ to the subgradient $\boldsymbol{g}_t$. The learner then receives the perturbed subgradient $\tilde{\boldsymbol{g}}_t = \boldsymbol{g}_t + \boldsymbol{\xi}_t$. We allow each $\boldsymbol{\xi}_t$ to follow a different distribution each round to satisfy different privacy guarantees. In the standard OCO framework the goal of the learner is to minimize the *regret* with respect to some parameter $\boldsymbol{u} \in \mathbb{R}^d$:

$$\mathcal{R}_T(\boldsymbol{u}) = \sum_{t=1}^{T} \ell_t(\boldsymbol{w}_t) - \ell_t(\boldsymbol{u}).$$

However, since the learner receives perturbed subgradients we consider the expected regret $\mathbb{E}[\mathcal{R}(\boldsymbol{u})]$, where the expectation is over the randomness in $\boldsymbol{w}_t$ due to the noisy subgradients. The setting will be formally introduced in section 2. Because $\tilde{\boldsymbol{g}}_t \in \mathbb{R}^d$, standard algorithms for unconstrained domains do not work since they require bounded $\tilde{\boldsymbol{g}}_t$. Initial work in this setting by Jun and Orabona (2019) was motivated by a lower bound of Cutkosky and Boahen (2017), which shows that one can suffer an exponential penalty when both the domain and subgradients are unbounded. They replace the boundedness assumption on $\tilde{\boldsymbol{g}}_t$ by a boundedness assumption on $\mathbb{E}[\tilde{\boldsymbol{g}}_t]$ and an assumption on the tails of the noise distribution. Jun and Orabona (2019) achieved expected regret guarantees of $O(\|\boldsymbol{u}\|\sqrt{(G^2 + \sigma^2)T \ln(1 + \|\boldsymbol{u}\|T)})$, where $\sigma^2$ is a uniform upper bound on $\mathbb{E}[\|\xi_t\|_\star^2]$, $G^2$ is a uniform upper bound on $\|\boldsymbol{g}_t\|_\star^2$, and $\|\cdot\|$ and $\|\cdot\|_\star$ are dual norms. This bound is useful when the distribution of the noise is constant and known and an adversary selects $\boldsymbol{g}_t$. We derive an algorithm that satisfies

$$E[\mathcal{R}_T(\boldsymbol{u})] = O\left(\|\boldsymbol{u}\|\sqrt{(G^2 T + \sum_{t=1}^{T} \sigma_t^2) \ln(1 + \|\boldsymbol{u}\|T))}\right), \tag{1}$$

where $\sigma_t^2 = \mathbb{E}[\|\xi_t\|_\star^2]$. This bound can be smaller in cases where only a few $\sigma_t$ are large but most are small. In fact, we will prove something stronger than (1):

$$\mathbb{E}[\mathcal{R}_T(\boldsymbol{u})] = O\left(\mathbb{E}[\|\boldsymbol{u}\|\sqrt{\sum_{t=1}^{T} \|\tilde{\boldsymbol{g}}_t\|_\star^2 \ln(1 + \|\boldsymbol{u}\|T))}]\right), \tag{2}$$

which implies (1) via Jensen's inequality and $\mathbb{E}[\|\tilde{\boldsymbol{g}}_t\|_\star^2] \leq 3\,\mathbb{E}[\|\boldsymbol{\xi}_t\|_\star^2] + 3\,\mathbb{E}[\|\boldsymbol{g}_t\|_\star^2]$. This bound was motivated by work in the noiseless setting, where $O(\|\boldsymbol{u}\|\sqrt{\sum_{t=1}^{T} \|\boldsymbol{g}_t\|_\star^2 \ln(1 + \|\boldsymbol{u}\|T)})$ bounds are possible (Cutkosky and Orabona, 2018). With these type of bounds, when the sum of the squared norms of the subgradients is small the regret is also small. To achieve (2) we require two assumptions: bounded $\|\boldsymbol{g}_t\|_\star$ and zero-mean symmetrical noise $\boldsymbol{\xi}_t$. The assumption on $\boldsymbol{g}_t$ is common in standard OCO. The symmetrical noise assumption is satisfied for common mechanisms to ensure local differential privacy. The dependence on $\mathbb{E}[\|\boldsymbol{\xi}_t\|_\star^2]$ and $\mathbb{E}[\|\boldsymbol{g}_t\|_\star^2]$ is unimprovable, which is shown by the lower bound for this setting by Jun and Orabona (2019).

The algorithms in this paper are built using the recently developed wealth-regret duality approach (Mcmahan and Streeter, 2012). We provide two algorithms. The first achieves the bound in (2). The second algorithm satisfies (2) for each dimension separately. This second algorithm can exploit sparse privacy structures, which combined with sparse subgradients yields low expected regret bounds.

**Contributions**   We extend the known results in several directions. Many common local differential privacy applications use symmetric additive noise (laplace mechanism, normal mechanism). We use the symmetry of the noise to adapt to unknown levels of privacy and achieve adaptive expected regret bounds. We also adapt to privacy for dimension specific dimension requirements, again without requiring knowledge of the structure of the noise other than symmetry in each dimension. Our algorithms interpolate between no noise and maximum noise, matching state of the art bounds in both cases. This can reduce the cost of privacy in some cases, outlined in section 4. Our work partially answers two problems left open by Jun and Orabona (2019). The first question asks whether or not data-dependent bounds are possible in the noisy OCO setting, which we answer affirmatively. The second question is how to adapt to different levels of noise without using extra parameters compared to the noiseless setting, which we do for symmetric noise.

**Related work**   There has been significant work on unconstrained and adaptive methods in OCO with noiseless subgradients $\boldsymbol{g}_t$ Foster et al. (2015); Orabona and Pál (2016); Foster et al. (2017); Cutkosky and Boahen (2017); Kotłowski (2017); Cutkosky and Orabona (2018); Foster et al. (2018). However, these results do not extend to the setting with noisy unbounded subgradients $\tilde{\boldsymbol{g}}_t$, which is possible with our work. For bounded domains regret bounds of $O(D\sqrt{\sum_{t=1}^{T}\|\tilde{\boldsymbol{g}}_t\|_{\star}^2})$ are possible without knowledge of the noise (Duchi et al., 2011; Orabona and Pál, 2018), where $D$ is an upper bound on $\|\boldsymbol{u}\|$. However, these bounds do not adapt to unknown $\|\boldsymbol{u}\|$, which may be costly for large $D$ but small $\|\boldsymbol{u}\|$. We provide an algorithm that both scales with $\|\boldsymbol{u}\|$ instead of $D$ and does not require knowledge of the noise.

There is a body of literature in the differential privacy setting with online feedback (Jain et al., 2012; Jain and Thakurta, 2014; Thakurta and Smith, 2013; Agarwal and Singh, 2017; Abernethy et al., 2017). In this paper we consider *local* differential privacy (Wasserman and Zhou, 2010; Duchi et al., 2014), which is a stronger notion of privacy than differential privacy. Duchi et al. (2014) provide an algorithm with constant local differential privacy that learns by using SGD. (Song et al., 2015) derive how to use knowledge of several levels of local differential privacy for SGD, but only with two different levels of noise. Jun and Orabona (2019) consider local privacy with an unbounded domain and constant noise. With knowledge of the noise it is possible to extend the results of Jun and Orabona (2019) to achieve (1), but not (2).

**Outline**   In section 2 we introduce our problem formally and introduce the key techniques. In section 3 we derive a one-dimensional algorithm that achieves our goals, which we use in a black-box reduction in section 3.1 and we apply it coordinate-wise in section 3.2. Section 4 contains two scenarios in which our new algorithm achieves improvements compared to current algorithms. Finally, in section 5 we present our conclusions.

## 2   Problem Formulation and Preliminaries

In this section we describe our notation, introduce the version of local differential privacy we use, briefly introduce the OCO setting with noisy subgradients, and provide some background to the reward-regret duality paradigm.

**Notation.**   Random variable $x$ is called symmetric if the density function $\rho$ of the random variable $\boldsymbol{z} = \boldsymbol{x} - \mathbb{E}[\boldsymbol{x}]$ satisfies $\rho(\boldsymbol{z}) = \rho(-\boldsymbol{z})$. The inner product between vectors $\boldsymbol{g} \in \mathbb{R}^d$ and $\boldsymbol{w} \in \mathbb{R}^d$ is denoted by $\langle \boldsymbol{w}, \boldsymbol{g} \rangle$. The Fenchel conjugate of a convex function $F$, $F^{\star}$ is defined as $F^{\star}(\boldsymbol{w}) = \sup_{\boldsymbol{g}} \langle \boldsymbol{w}, \boldsymbol{g} \rangle - F(\boldsymbol{g})$. $\| \cdot \|$ denotes a norm and $\|\boldsymbol{g}\|_{\star} = \sup_{\boldsymbol{w}:\|\boldsymbol{w}\|\leq 1} \langle \boldsymbol{w}, \boldsymbol{g} \rangle$ denotes the dual norm. $g_{t,j}$ indicates the $j^{\text{th}}$ component of vector $\boldsymbol{g}_t$.

### 2.1   User-Specified Local Differential Privacy

In the local differential privacy setting each datum is kept private from the learner. The standard definition of local privacy requires a randomiser $R$ that perturbs $\boldsymbol{g}_t$ with random noise $\boldsymbol{\xi}_t$, where $\boldsymbol{\xi}_1, \ldots, \boldsymbol{\xi}_T$ are independently distributed (Wasserman and Zhou, 2010; Kasiviswanathan et al., 2011; Duchi et al., 2014). The amount of perturbation is controlled by $\epsilon$, where smaller $\epsilon$ means more privacy. We allow the provider to specify his desired level of privacy, so in a given round $t$ we have $\epsilon_t$-local differential privacy.

**Definition 1.** *[Duchi et al. (2014)] Let $A = (X_1, \ldots, X_T)$ be a sensitive dataset where each $X_t \in A$ corresponds to data about individual $t$. A randomiser $R$ which outputs a disguised version of $S = (U_1, \ldots, U_T)$ of $A$ is said to provide $\epsilon$-local differential privacy to individual $t$, if for all $x, x' \in A$ and for all $S \subseteq \mathcal{S}$,*

$$\Pr(U_t \in S | X_t = x) \leq \exp(\epsilon) \Pr(U_t \in S | X_t = x').$$

In this paper we make use of randomisers of the form $R_t(\boldsymbol{g}_t) = \boldsymbol{g}_t + \boldsymbol{\xi}_t$, where $\boldsymbol{\xi}_t$ is generated by a zero-mean symmetrical distribution $\rho_t$. A common choice for $\rho_t$ is $\rho_t(\boldsymbol{z}) \propto \exp(-\frac{\epsilon_t}{2}\|\boldsymbol{z}\|)$ (Song et al., 2015). This randomiser is $\epsilon_t$-local differentially private for $\|\boldsymbol{g}_t\| \leq 1$ (Song et al., 2015, Theorem 1). We make use a small variation of this randomiser, which we call the local Laplace randomiser: $\rho_t(\boldsymbol{z}) \propto \exp(-\sum_{j=1}^{d} \frac{\tau_{t,j}}{2}|\boldsymbol{z}_j|)$, where $\sum_{j=1}^{d} \tau_{t,j} = \epsilon_t$, $\tau_{t,j} \geq 0$. The following result shows that the local Laplace randomiser preserves $\epsilon_t$-local differential privacy.

**Lemma 1.** *Suppose $|g_{t,j}| \leq 1$, then the local Laplace randomiser is $\epsilon_t$-local differentially private, where $\epsilon_t = \sum_{j=1}^{d} \tau_{t,j}$.*

The proof follows from applying Theorem 1 of Song et al. (2015) to each dimension and summing the $\tau_{t,j}$. For completeness the proof is provided in Appendix A. This randomiser is the Laplace randomiser (Dwork and Roth, 2014) applied to each dimension with a possibly different $\epsilon$ per dimension. The local Laplace randomiser gives the user more control over the details of the privacy guarantees: with the local Laplace randomiser each dimension $j$ is $\tau_{t,j}$-local differentially private. This can also lead to lower regret in some cases, of which we give an example in section 4.

## 2.2   Online Convex Optimization with Noisy Subgradients

The analysis of many efficient online learning tools has been influenced by the Online Convex Optimization framework. As mentioned in the introduction, the OCO setting with noisy subgradients proceeds in rounds $t = 1, \ldots, T$. In each round $t$

1. The learner sends $\boldsymbol{w}_t \in \mathbb{R}^d$ to the provider of the $t^{\text{th}}$ subgradient.
2. The provider samples $\boldsymbol{\xi}_t$ from zero-mean and symmetrical $\rho_t$ and computes subgradient $\boldsymbol{g}_t \in \partial \ell_t(\boldsymbol{w}_t)$, where $\|\boldsymbol{g}_t\|_\star \leq G$.
3. The provider sends $\tilde{\boldsymbol{g}}_t = \boldsymbol{g}_t + \boldsymbol{\xi}_t \in \mathbb{R}^d$ to the learner.

This protocol is a slight adaptation of the protocol of Duchi et al. (2014), where we allow a different $\rho_t$ in each round $t$ instead of using a constant $\rho$. In each round the provider only sends $\tilde{\boldsymbol{g}}_t$ to the learner. The learner has no information about $\rho_t$ other than that $\rho_t$ is symmetrical and zero-mean. Also note that $\rho_t$ is allowed to change with each round $t$, complicating things even further. Since the feedback the learner receives is random we are interested in the expected regret. To bound the expected regret we upper bound the losses by their tangents:

$$\mathbb{E}[\mathcal{R}_T(\boldsymbol{u})] \leq \mathbb{E}[\sum_{t=1}^{T} \langle \boldsymbol{w}_t - \boldsymbol{u}, \boldsymbol{g}_t \rangle] = \mathbb{E}[\sum_{t=1}^{T} \langle \boldsymbol{w}_t - \boldsymbol{u}, \tilde{\boldsymbol{g}}_t \rangle], \tag{3}$$

where the equality holds because of the law of total expectation. The analysis focusses on bounding the r.h.s of (3), which is a standard approach in OCO. In the following we introduce a recently popularized method to control the regret when $\boldsymbol{w}_t$ and $\boldsymbol{u}$ are unbounded.

## 2.3   Reward Regret Duality

For noisy $\tilde{\boldsymbol{g}}_t$, the formal result is found in the following lemma (see also Theorem 3 of Jun and Orabona (2019)).

**Lemma 2.** *If $-\mathbb{E}[\sum_{t=1}^{T} \langle \boldsymbol{w}_t, \boldsymbol{g}_t \rangle] \geq \mathbb{E}[F_T(-\sum_{t=1}^{T} \tilde{\boldsymbol{g}}_t) - c_T]$ for some convex function $F_T$ and $c_T \in \mathbb{R}$, then $\mathbb{E}[\mathcal{R}_T(\boldsymbol{u})] \leq E[c_T] + F_T^\star(\boldsymbol{u})$.*

*Proof.* From the definition of Fenchel conjugates we have $\mathbb{E}[F_T(-\sum_{t=1}^{T} \tilde{\boldsymbol{g}}_t)] \geq \mathbb{E}[-F_T^\star(\boldsymbol{u}) - \sum_{t=1}^{T} \langle \boldsymbol{u}, \tilde{\boldsymbol{g}}_t \rangle] = -F_T^\star(\boldsymbol{u}) - \sum_{t=1}^{T} \langle \boldsymbol{u}, \boldsymbol{g}_t \rangle$. Using $-\mathbb{E}[\sum_{t=1}^{T} \langle \boldsymbol{w}_t, \boldsymbol{g}_t \rangle] \geq \mathbb{E}[F_T(-\sum_{t=1}^{T} \tilde{\boldsymbol{g}}_t) - c_T]$ and reordering the terms completes the proof. $\square$

The difficulty lies in finding a suitable $F_T$ and $c_T$. For example, we could use gradient descent with learning rate $\eta$ to find $F_T(-\sum_{t=1}^{T} \tilde{g}_t) = \frac{\eta}{2}\|\sum_{t=1}^{T} \tilde{g}_t\|_2^2$ and $c_T = \sum_{t=1}^{T} \frac{\eta}{2}\|\tilde{g}_t\|_2^2$. However, it would be impossible to tune $\eta$ optimally due to the dependence on the unknown $\boldsymbol{u}$ in $F_T^*(\boldsymbol{u}) = \frac{1}{2\eta}\|\boldsymbol{u}\|_2^2$. For noiseless subgradients $\boldsymbol{g}_t$ (Cutkosky and Orabona, 2018) provide a route to find a suitable $F_T$, with a constant $c_T$. Jun and Orabona (2019) extend this idea to noisy subgradients $\tilde{\boldsymbol{g}}_t$: one needs to find an $F_t$, $F_{t-1}$, and $\boldsymbol{w}_t$ that satisfy $F_{t-1}(\boldsymbol{x}) - \langle \boldsymbol{w}_t, \boldsymbol{g}_t \rangle \geq \mathbb{E}_{\tilde{\boldsymbol{g}}_t}[F_t(\boldsymbol{x} - \tilde{\boldsymbol{g}}_t)]$. By assuming that $-\mathbb{E}[\sum_{s=1}^{t} \langle \boldsymbol{w}_s, \boldsymbol{g}_s \rangle] \geq \mathbb{E}[F_t(-\sum_{s=1}^{t} \tilde{\boldsymbol{g}}_s)]$ holds one can show that if $F_t$ and $F_{t-1}$ satisfy $F_{t-1}(\boldsymbol{x}) - \langle \boldsymbol{w}_t, \boldsymbol{g}_t \rangle \geq \mathbb{E}_{\tilde{\boldsymbol{g}}_t}[F_t(\boldsymbol{x} - \tilde{\boldsymbol{g}}_t)]$, $-\mathbb{E}[\sum_{t=1}^{T} \langle \boldsymbol{w}_t, \boldsymbol{g}_t \rangle] \geq \mathbb{E}[F_T(-\sum_{t=1}^{T} \tilde{\boldsymbol{g}}_t)]$ holds by induction. The result is given in the following lemma, of which the proof can be found in Appendix A.

**Lemma 3.** *Suppose that* $F_{t-1}(\boldsymbol{x}) - \langle \boldsymbol{w}_t, \boldsymbol{g}_t \rangle \geq \mathbb{E}_{\tilde{\boldsymbol{g}}_t}[F_t(\boldsymbol{x} - \tilde{\boldsymbol{g}}_t)]$ *holds for all* $t$, *then*

$$-\mathbb{E}[\sum_{t=1}^{T} \langle \boldsymbol{w}_t, \boldsymbol{g}_t \rangle] \geq \mathbb{E}[F_T(-\sum_{t=1}^{T} \tilde{\boldsymbol{g}}_t)].$$

## 3 One-Dimensional Private Adaptive Potential Function

---
**Algorithm 1** Local Differentially Private Adaptive Potential Function

---
**Require:** $G$ such that $|\mathbb{E}[\tilde{g}_t]| \leq G$ and prior $P$ on $v \in [-\frac{1}{5G}, \frac{1}{5G}]$.
1: **for** $t = 1, \ldots, T$ **do**
2:      Play $w_t = \mathbb{E}_{v \sim P}[v \exp(-\sum_{s=1}^{t-1} v\tilde{g}_s - (v\tilde{g}_s)^2)]$.
3:      Receive symmetric $\tilde{g}_t \in \mathbb{R}$.
4: **end for**

---

In this section we derive a suitable potential function for a one-dimensional problem. In the remainder of this paper we use this one-dimensional potential to derive new algorithms. To derive our one-dimensional potential function we we rely on a property of symmetric random variables with bounded means. The following Lemma is key deriving our potential function $F_T$.

**Lemma 4.** *Suppose* $\boldsymbol{x}$ *is a symmetrical random variable with* $|\mathbb{E}[\langle \boldsymbol{v}, \boldsymbol{x} \rangle]| \leq \frac{1}{5}$ *for some* $\boldsymbol{v}$. *Then* $\mathbb{E}[\exp(\langle \boldsymbol{v}, \boldsymbol{x} \rangle - \langle \boldsymbol{v}, \boldsymbol{x} \rangle^2)] \leq 1 + \mathbb{E}[\langle \boldsymbol{v}, \boldsymbol{x} \rangle]$.

The proof of Lemma 4 can be found in Appendix B. We can now use Lemma 4 to derive a one-dimensional potential function. Suppose $\tilde{g}_t \in \mathbb{R}$ is a symmetrical random variable with $\mathbb{E}[\tilde{g}_t] \leq G$. Then $v\tilde{g}_t$ with $v \in [-\frac{1}{5G}, \frac{1}{5G}]$ satisfies the assumptions in Lemma 4. Multiplying the lower bound of Lemma 4 for $1 - \mathbb{E}[v\tilde{g}_t]$, for $t = 1, \ldots, T$, yields a potential function via Lemma 3. The potential we find is

$$\mathbb{E}[F_t(-\sum_{s=1}^{t} \tilde{g}_s)] = \mathbb{E}[\underset{v \sim P}{\mathbb{E}}[\exp(-\sum_{s=1}^{t} v\tilde{g}_s - (v\tilde{g}_s)^2) - 1]], \tag{4}$$

where $P$ is an (improper) prior on $v \in [-\frac{1}{5G}, \frac{1}{5G}]$, the first expectation is over $\tilde{g}_1, \ldots, \tilde{g}_t$, and $F_0(0) = 0$. This kind of potential function has been used before by Chernov and Vovk (2010); Koolen and Van Erven (2015); Jun and Orabona (2019). The novelty in this particular potential function is the incorporation of the symmetrical noise. The $\sum_{s=1}^{t}(v\tilde{g}_s)^2$ term is unique to our potential function and allows us to derive adaptive regret bounds for unconstrained $u$. Note that the $c_T = 1$ term has moved inside the definition of $F_T$. While this does not influence the analysis for proper priors it does influence the analysis for improper priors. The corresponding prediction strategy is given by

$$w_t = \underset{v \sim P}{\mathbb{E}}[v \exp(-\sum_{s=1}^{t-1} v\tilde{g}_s - (v\tilde{g}_s)^2)]. \tag{5}$$

Algorithm 1 summarizes the strategy. Note that Algorithm 1 does not require any extra parameters compared to the setting with noiseless subgradients.

The following result shows that $F_T$ defined by (4) and $w_t$ defined by (5) satisfy our assumptions.

**Lemma 5.** *Suppose* $\tilde{g}_t$ *is a symmetrical random variable with* $\mathbb{E}[\tilde{g}_t] \leq G$. *Then* $F_t$ *defined by* (4) *and* $w_t$ *defined by* (5) *satisfy* $\mathbb{E}_{\tilde{g}_t}[F_t(-\sum_{s=1}^{t} \tilde{g}_s)] \leq F_{t-1}(-\sum_{s=1}^{t-1} \tilde{g}_s) - w_t \mathbb{E}[\tilde{g}_t]$.

The proof follows from an application of Lemma 4 and can be found in Appendix B. We consider two types of priors. The first type are proper priors that are of the form:

$$\frac{dP(v)}{dv} = \frac{\nu(v)\exp(-bv^2)}{Z},\tag{6}$$

Where $b \geq 0$, $\nu : [-\frac{1}{5G}, \frac{1}{5G}] \mapsto \mathbb{R}_+$, and $Z = \int_{-\frac{1}{5G}}^{\frac{1}{5G}} \nu(v)e^{-bv^2} dv$ is a normalizing constant. This captures several priors used in literature, including the conjugate prior $\frac{dP}{dv} = \frac{\exp(-bv^2)}{Z}$ (Koolen and Van Erven, 2015), a variant of the *CV* prior $\frac{dP}{dv} = \frac{1}{Z|v|\ln(|v|)^2}$ (for $G > \frac{1}{5}$), (Chernov and Vovk, 2010; Koolen and Van Erven, 2015), and the uniform prior on $[-\frac{1}{5G}, \frac{1}{5G}]$ (Jun and Orabona, 2019).

The second type of prior is an improper prior: $\frac{dP}{dv} = \frac{1}{|v|}$. A variant of this prior was previously used by (Koolen and Van Erven, 2015). For all priors we derive a regret bound by computing an upper bound on the convex conjugate of $F_T$, $F_T^\star$. For conciseness we only present the regret bound for the conjugate prior in the main text. In Appendix C we present the analysis of the regret of the improper prior, for which a slightly different analysis is required compared to the proper priors. The analysis for all priors can be seen as performing a Laplace approximation of the integral over $v$ to show that the prior places sufficient mass in a neighbourhood of the optimal $v$.

Abbreviating $B_t = b + \sum_{s=1}^{t-1} \tilde{g}_s^2$, $L_t = -\sum_{s=1}^{t-1} \tilde{g}_s$, and $C = \frac{1}{5G}$, the predictions (5) with the conjugate prior are given by:

$$w_t = \frac{\sqrt{b}L_t \exp\left(\frac{(L+2CB_t)^2}{4B_t}\right)\left(\mathrm{erf}\left(\frac{L_t-2CB_t}{2\sqrt{B_t}}\right) - \mathrm{erf}\left(\frac{L_t+2CB_t}{2\sqrt{B_t}}\right)\right) + 2\sqrt{B_t}\left(\exp(2CL_t)-1\right)}{\mathrm{erf}(C\sqrt{b})\exp(C\left(L_t+CB_t\right))4B_t^{\frac{3}{2}}}.\tag{7}$$

These $w_t$ can be computed efficiently, but see Koolen and Van Erven (2015) for numerically stable evaluation. With the conjugate prior we find the following result:

**Theorem 1.** *Suppose $\tilde{g}_t$ is a symmetrical random variable with $|\mathbb{E}[\tilde{g}_t]| \leq G$ for all t. The predictions* (7) *satisfy:*

$$\mathbb{E}[\mathcal{R}_T(u)] \leq 1 + |u| \max\left\{ 11G\left(\ln(|u|11G) - 1 + \ln\left(\frac{\sqrt{5}G\sqrt{\pi}}{4\sqrt{b}}\right)\right), \right.$$
$$\left. \mathbb{E}\left[\sqrt{8\left(b + \sum_{t=1}^{T}\tilde{g}_t^2\right)\ln(16|u|^2\left(b + \sum_{t=1}^{T}\tilde{g}_t^2\right)^{\frac{3}{2}}\frac{\sqrt{\pi}}{\sqrt{b}} + 1)}\right]\right\}.$$

The proof of Theorem 1 can be found in Appendix B.1 and follows from computing the Fenchel conjugate of the potential function. For noisy subgradients this is the first bound that is adaptive to the sum of the squares of the noisy subgradients. Compared to the expected regret bound for the improper prior (see Theorem 3 in Appendix C) this bound has worse constants. However, with the conjugate prior all non-constant terms scale with $|u|$, which is not the case with the improper prior. For all proper priors of the form (6) a similar regret bound can be computed. This can be seen from Lemma 8 in Appendix B.1, which shows that the convex conjugate of the potential function for these priors is $O(\mathbb{E}[|u|\sqrt{\sum_{t=1}^{T}\tilde{g}_t^2 \ln(|u|T + 1))}])$.

## 3.1 Black-Box Reductions

In this section we use our potential function in a black-box reduction: we take a constrained noisy OCO algorithm $\mathcal{A}_{\mathcal{Z}}$ and turn it into an unconstrained algorithm using our potential function. The same reduction is used by Cutkosky and Orabona (2018) and Jun and Orabona (2019). The algorithm can be found in Figure 2. The potential function and the OCO algorithm each have their task: the potential function is to learn the norm of $\boldsymbol{u}$ and the constrained OCO algorithm is to learn the direction of $\boldsymbol{u}$. In each round $t$ we play $\boldsymbol{w}_t = v_t \boldsymbol{z}_t$, where $\boldsymbol{z}_t \in \mathcal{Z}$, $\mathcal{Z} = \{\boldsymbol{z} : \|\boldsymbol{z}\| \leq 1\}$, is the prediction of the OCO algorithm and $v_t$ is the prediction of Algorithm 1. We feed $\tilde{\boldsymbol{g}}_t$ as feedback to $\mathcal{A}_{\mathcal{Z}}$ and $\langle \boldsymbol{z}_t, \tilde{\boldsymbol{g}}_t \rangle$ as feedback to Algorithm 1. Since $\tilde{\boldsymbol{g}}_t$ is a symmetrical random variable and $\mathbb{E}[\langle \boldsymbol{z}_t, \tilde{\boldsymbol{g}}_t \rangle] \leq G$,

---

**Algorithm 2** Black-Box Reduction

---

**Require:** $G$ such that $\|\mathbb{E}[\tilde{\boldsymbol{g}}_t]\|_\star \leq G$ and Algorithm $\mathcal{A}_\mathcal{Z}$ with domain $\mathcal{Z} = \{\boldsymbol{z} : \|\boldsymbol{z}\| \leq 1\}$
 1: **for** $t = 1, \ldots, T$ **do**
 2:     Get $\boldsymbol{z}_t \in \mathcal{Z}$ from $\mathcal{A}_\mathcal{Z}$
 3:     Get $v_t \in \mathbb{R}$ from Algorithm 1
 4:     Play $\boldsymbol{w}_t = v_t \boldsymbol{z}_t$, receive symmetrical $\tilde{\boldsymbol{g}}_t$ such that $\|\mathbb{E}[\tilde{\boldsymbol{g}}_t]\|_\star \leq G$
 5:     Send $\tilde{\boldsymbol{g}}_t$ to $\mathcal{A}_\mathcal{Z}$
 6:     Send $\langle \boldsymbol{z}_t, \tilde{\boldsymbol{g}}_t \rangle$ to Algorithm 1
 7: **end for**

---

$\langle \boldsymbol{z}_t, \tilde{\boldsymbol{g}}_t \rangle$ satisfies the assumptions in Lemma 4. This allows us to control the regret for learning the norm of $\boldsymbol{u}$ using Theorem 1.

As outlined by Cutkosky and Orabona (2018) the expected regret of Algorithm 2 decomposes into two parts. The first part of the regret is for learning the norm of $\boldsymbol{u}$, and is controlled by Algorithm 1. The second part of the regret for learning the direction of $\boldsymbol{u}$ and is controlled by $\mathcal{A}_\mathcal{Z}$. The proof is given by Cutkosky and Orabona (2018), but for completeness we provide the proof in Appendix B.2.

**Lemma 6.** *Suppose $\tilde{\boldsymbol{g}}_t$ is a symmetrical random variable with $\|\mathbb{E}[\tilde{\boldsymbol{g}}_t]\|_\star \leq G$ for all $t$. Let $\mathcal{R}_T^\mathcal{V}(\|\boldsymbol{u}\|) = \mathbb{E}[\sum_{t=1}^T (v_t - \|\boldsymbol{u}\|)\langle \boldsymbol{z}_t, \tilde{\boldsymbol{g}}_t \rangle]$ be the regret for learning $\|\boldsymbol{u}\|$ by Algorithm 1 and let $\mathcal{R}_T^\mathcal{Z}(\frac{\boldsymbol{u}}{\|\boldsymbol{u}\|}) = \mathbb{E}[\sum_{t=1}^T \langle \boldsymbol{z}_t - \frac{\boldsymbol{u}}{\|\boldsymbol{u}\|}, \tilde{\boldsymbol{g}}_t \rangle]$ be the regret for learning $\frac{\boldsymbol{u}}{\|\boldsymbol{u}\|}$ by $\mathcal{A}_\mathcal{Z}$. Then Algorithm 2 satisfies $\mathbb{E}[\mathcal{R}_T(\boldsymbol{u})] = \mathcal{R}_T^\mathcal{V}(\|\boldsymbol{u}\|) + \|\boldsymbol{u}\|\mathcal{R}_T^\mathcal{Z}\left(\frac{\boldsymbol{u}}{\|\boldsymbol{u}\|}\right).$*

Orabona and Pál (2018) show that Mirror Descent with learning rates $\eta_t = (\sqrt{\sum_{s=1}^t \|\tilde{\boldsymbol{g}}_s\|_\star^2})^{-1}$ yields $\mathcal{R}_T^\mathcal{Z}(\frac{\boldsymbol{u}}{\|\boldsymbol{u}\|}) = O(\mathbb{E}[\sqrt{\sum_{t=1}^T \|\tilde{\boldsymbol{g}}_t\|_\star^2}])$. Since Algorithm 1 satisfies $\mathcal{R}_T^\mathcal{V}(\|\boldsymbol{u}\|) = O(\mathbb{E}[\|\boldsymbol{u}\|\sqrt{\sum_{t=1}^T \|\tilde{\boldsymbol{g}}_t\|_\star^2 \ln(\|\boldsymbol{u}\|\sum_{t=1}^T \|\tilde{\boldsymbol{g}}_t\|_\star^2 + 1)}])$ the total regret of Algorithm 2 is

$$\mathbb{E}[\mathcal{R}_T(\boldsymbol{u})] = O\left(\|\boldsymbol{u}\|\,\mathbb{E}\left[\sqrt{\sum_{t=1}^T \|\tilde{\boldsymbol{g}}_t\|_\star^2 \ln(\|\boldsymbol{u}\|\sum_{t=1}^T \|\tilde{\boldsymbol{g}}_t\|_\star^2 + 1)}\right]\right). \tag{8}$$

This bound matches state of the art bounds for for noiseless subgradients and is never worse than the bound of Jun and Orabona (2019) for noisy subgradients, but can be substantially better.

### 3.2   Private Unconstrained Adaptive Sparse Gradient Descent

---

**Algorithm 3** Private Unconstrained Adaptive Sparse Gradient Descent

---

**Require:** $G$ such that $|\mathbb{E}[\tilde{g}_{t,j}]|_\star \leq G$.
 1: **for** $t = 1, \ldots, T$ **do**
 2:     Play $\boldsymbol{w}_t$
 3:     **for** $j = 1, \ldots, d$ **do**
 4:         Receive symmetrical $\tilde{g}_{t,j}$ such that $|\tilde{g}_{t,j}| \leq G$
 5:         Send $\tilde{g}_{t,j}$ to Algorithm 1
 6:         Receive $v_{t+1} \in \mathbb{R}$ from Algorithm 1 with the conjugate prior
 7:         Set $\boldsymbol{w}_{t+1,j} = v_{t+1}$
 8:     **end for**
 9: **end for**

---

In this section we propose a noisy unconstrained OCO algorithm that can exploit sparse subgradients. The algorithm is summarized in Algorithm 3. Algorithm 3 runs a copy of Algorithm 1 with the conjugate prior coordinate-wise. A similar strategy is used by Orabona and Tommasi (2017). This strategy can exploit sparse privacy structures, which, combined with sparse subgradients, may yield low regret (see section 4). Its expected regret bound is given below. The proof follows from applying Theorem 1 per dimension.

**Theorem 2.** *Suppose $\tilde{g}_{t,j}$ is a symmetric random variable with $|\mathbb{E}[\tilde{g}_{t,j}]| \leq G$ for all $t$ and $j$. Then the expected regret of Algorithm 3 satisfies*

$$\mathbb{E}[\mathcal{R}_T(u)] \leq d + \sum_{j=1}^{d} |\boldsymbol{u}_j| \max \left\{ 11G \left( \ln(|\boldsymbol{u}_j|11G) - 1 + \ln \left( \frac{\sqrt{5}G\sqrt{\pi}}{4\sqrt{b_j}} \right) \right), \right.$$

$$\left. \mathbb{E}\left[ \sqrt{ 8 \left( b_j + \sum_{t=1}^{T} \tilde{g}_{t,j}^2 \right) \ln(16|\boldsymbol{u}_j|^2 \left( b_j + \sum_{t=1}^{T} \tilde{g}_{t,j}^2 \right)^{\frac{3}{2}} \frac{\sqrt{\pi}}{\sqrt{b_j}} + 1)) } \right] \right\}.$$

## 4 Motivating Examples

In this section we present two scenarios in which our algorithms provide better expected regret guarantees than standard algorithms. The first scenario concerns a case where many providers do not care for their privacy (so they do not perturb the subgradients) and few providers care substantially for their privacy. Suppose that the providers who care for their privacy are $\lceil \ln(T) \rceil$ of the total number of providers $T$. Suppose that $\|\boldsymbol{g}_t\|_2^2 \leq 1$ and that the providers who care for their privacy use $\rho(\boldsymbol{z}) \propto \exp(-\frac{\epsilon}{2}\|\boldsymbol{z}\|_2)$, then $\mathbb{E}[\|\boldsymbol{\xi}_t\|_2^2] \leq 4 + 4\frac{d^2+d}{\epsilon^2}$ (Song et al., 2015, Theorem 1). Using Algorithm 2, Jensen's inequality, and the fact that the square root is subadditive we see from (8) that the expected regret is upper bounded by $O(\|\boldsymbol{u}\|_2\sqrt{\sum_{t=1}^{T}\|\boldsymbol{g}_t\|_2^2\ln(1+\|\boldsymbol{u}\|_2T)} + \|\boldsymbol{u}\|_2\frac{d}{\epsilon}\ln(\|\boldsymbol{u}\|_2T+T))$ instead of $O(\|\boldsymbol{u}\|_2\frac{d}{\epsilon}\sqrt{T\ln(1+\|\boldsymbol{u}\|_2T)})$ had we used the maximum privacy guarantee for all providers instead of letting the providers choose their desired level of privacy.

In the second scenario the providers use the local Laplace randomiser. Suppose that $\boldsymbol{g}_t$ is sparse. A standard algorithm that has good performance for sparse $\boldsymbol{g}_t$ is AdaGrad (Duchi et al., 2011). AdaGrad achieves $O(\mathbb{E}[D\sum_{j=1}^{d}\sqrt{\sum_{t=1}^{T}|\tilde{g}_{t,j}|^2}])$ expected regret, where $\max_j |\boldsymbol{u}_j| \leq D$, and $D$ has to be guessed prior to running AdaGrad. Using Jensen's inequality and the fact that the square root is subadditive the expected regret can be upper bounded by $O(D\sum_{j=1}^{d}(\sqrt{3\sum_{t=1}^{T}\mathbb{E}[\boldsymbol{g}_{t,j}^2]} + \sqrt{\sum_{t=1}^{T}3\,\mathbb{E}[\boldsymbol{\xi}_{t,j}^2]}))$. Algorithm 3 achieves $O(\sum_{j=1}^{d}|\boldsymbol{u}_j|(\sqrt{3\sum_{t=1}^{T}\mathbb{E}[\boldsymbol{g}_{t,j}^2]\ln(|\boldsymbol{u}_j|T+1)} + \sqrt{3\sum_{t=1}^{T}\mathbb{E}[\boldsymbol{\xi}_{t,j}^2]\ln(|\boldsymbol{u}_j|T+1)}))$ regret, which can be significantly smaller than the bound of AdaGrad if $D$ is much larger than $\boldsymbol{u}_j$ or if $\boldsymbol{u}$ is sparse. Furthermore, since we allow the provider of the data to choose $\tau_{t,j}$, $\boldsymbol{\xi}_t$ can be sparse as well. While this does not give local differential privacy guarantees for all attributes it does give local differential privacy guarantees for attributes with $\tau_j < \infty$. If instead we would have used the standard application of local differential privacy there would be no hope in exploiting sparse $\boldsymbol{g}_t$ since $\sum_{j=1}^{d}|\boldsymbol{u}_j|\sqrt{3\sum_{t=1}^{T}\mathbb{E}[\boldsymbol{\xi}_{t,j}^2]}$ would be the dominant term in the regret bound.

## 5 Conclusions

In this paper, we extended the local differential privacy framework in unconstrained Online Convex Optimization by allowing the provider of the data to choose their privacy guarantees. Standard algorithms do not yield satisfactory regret bounds in this setting, either due to dependence on the unknown parameters of the noise or dependence on bounded subgradients. Hence, we proposed two new algorithms that match state of the art regret algorithms in both the noisy and noiseless setting, without requiring knowledge of the noise other than symmetry. Our algorithms do not require parameters other than a bound on the norm of expectation of the subgradients, which allows the privacy requirements of all providers to be private itself. The new algorithms are a step towards practically useful algorithms with local differential privacy guarantees that have sound theoretical guarantees. Our algorithms are the first adaptive unconstrained algorithms in the noisy OCO setting without requiring extra parameters compared to the standard OCO setting, solving two problems left open by Jun and Orabona (2019).

**Acknowledgments**

The author would like to thank Tim van Erven for his comments on an earlier version of this paper. The author was supported by the Netherlands Organization for Scientific Research (NWO grant TOP2EW.15.211).

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
