[Supplementary Material · AdaPrivate_supp_final.pdf]

# A   Details from Section 2

*Proof.* (of Lemma 1) Evaluating and rewriting Definition 1 gives

$$\prod_{j=1}^{d} \frac{\exp(-\frac{\tau_{t,j}}{2}|\tilde{g}_{t,j} - \boldsymbol{g}_{t,j}|)}{\exp(-\frac{\tau_{t,j}}{2}|\tilde{g}_{t,j} - \boldsymbol{g}'_{t,j}|)} \leq \prod_{j=1}^{d} \exp(\frac{\tau_{t,j}}{2}(|\boldsymbol{g}_{t,j}| + |\boldsymbol{g}'_{t,j}|)) \leq \prod_{j=1}^{d} \exp(\tau_{t,j}) = \exp(\epsilon_t),$$

where the first inequality follows from applying the triangle inequality for each $j$ and the second inequality follows from the assumption that $|\boldsymbol{g}_{t,j}| \leq 1$. □

*Proof.* (of Lemma 3) We will prove the result by induction. In a given round $t$ assume that $-\mathbb{E}[\sum_{s=1}^{t}\langle\boldsymbol{w}_s, \boldsymbol{g}_s\rangle] \geq \mathbb{E}[F_t(-\sum_{s=1}^{t}\tilde{\boldsymbol{g}}_s)]$ holds. Now,

$$\begin{aligned}
-\mathbb{E}[\sum_{s=1}^{t+1}\langle\boldsymbol{w}_s, \boldsymbol{g}_s\rangle] &= \mathbb{E}[-\langle\boldsymbol{w}_{t+1}, \boldsymbol{g}_{t+1}\rangle - \sum_{s=1}^{t}\langle\boldsymbol{w}_s, \boldsymbol{g}_s\rangle] \\
&\geq \mathbb{E}[F_t(-\sum_{s=1}^{t}\tilde{\boldsymbol{g}}_s) - \langle\boldsymbol{w}_{t+1}, \boldsymbol{g}_{t+1}\rangle] \\
&\geq \mathbb{E}[F_{t+1}(-\sum_{s=1}^{t+1}\tilde{\boldsymbol{g}}_s)],
\end{aligned}$$

where the first inequality comes from the inductive hypothesis and the second inequality is by the assumption that $F_{t-1}(\boldsymbol{x}) - \langle\boldsymbol{w}_t, \boldsymbol{g}_t\rangle \geq \mathbb{E}_{\tilde{\boldsymbol{g}}_t}[F_t(\boldsymbol{x} - \tilde{\boldsymbol{g}}_t)]$ for all $t$. Now, by induction $-\mathbb{E}[\sum_{t=1}^{T}\langle\boldsymbol{w}_t, \boldsymbol{g}_t\rangle] \geq \mathbb{E}[F_T(-\sum_{t=1}^{T}\tilde{\boldsymbol{g}}_t)]$. □

# B   Details from Section 3

*Proof.* (of Lemma 4) We start by rewriting the l.h.s.:

$$\mathbb{E}[\exp(\langle\boldsymbol{v}, \boldsymbol{x}\rangle - \langle\boldsymbol{v}, \boldsymbol{x}\rangle^2)] = \mathbb{E}[\exp(y\langle\boldsymbol{v}, \boldsymbol{z}\rangle - \langle\boldsymbol{v}, \boldsymbol{z}\rangle^2)]\exp(\mathbb{E}[\langle\boldsymbol{v}, \boldsymbol{x}\rangle] - \mathbb{E}[\langle\boldsymbol{v}, \boldsymbol{x}\rangle]^2).$$

where $\boldsymbol{z} = \boldsymbol{x} - \mathbb{E}[\boldsymbol{x}]$ and $y = 1 - 2\mathbb{E}[\langle\boldsymbol{v}, \boldsymbol{x}\rangle]$. $\boldsymbol{z}$ is a random variable with mean 0 and $|y| \leq 1.4$ due to the restrictions on $\mathbb{E}[\langle\boldsymbol{v}, \boldsymbol{x}\rangle]$. By Lemma 7 $\mathbb{E}[\exp(y\langle\boldsymbol{v}, \boldsymbol{z}\rangle - \langle\boldsymbol{v}, \boldsymbol{z}\rangle^2)] \leq 1$. It remains to show that $\exp(\mathbb{E}[\langle\boldsymbol{v}, \boldsymbol{x}\rangle] - \mathbb{E}[\langle\boldsymbol{v}, \boldsymbol{x}\rangle]^2) \leq 1 + \mathbb{E}[\langle\boldsymbol{v}, \boldsymbol{x}\rangle]$, which holds for $\mathbb{E}[\langle\boldsymbol{v}, \boldsymbol{x}\rangle] \geq -\frac{1}{2}$ (Cesa-Bianchi and Lugosi, 2006, Lemma 2.4). □

**Lemma 7.** *Let $\boldsymbol{z} \in \mathbb{R}^d$ be a zero-mean symmetrical random variable. Then for $|y| \leq 1.4$ and arbitrary $\boldsymbol{v} \in \mathbb{R}^d$*

$$\mathbb{E}[\exp(y\langle\boldsymbol{v}, \boldsymbol{z}\rangle - \langle\boldsymbol{v}, \boldsymbol{z}\rangle^2)] \leq 1.$$

*Proof.* Due to symmetry of $\boldsymbol{z}$ we can write

$$\mathbb{E}[\exp(y\langle\boldsymbol{v}, \boldsymbol{z}\rangle - \langle\boldsymbol{v}, \boldsymbol{z}\rangle^2)] = \mathbb{E}[\frac{1}{2}\exp(-y\langle\boldsymbol{v}, \boldsymbol{z}\rangle - \langle\boldsymbol{v}, \boldsymbol{z}\rangle^2) + \frac{1}{2}\exp(y\langle\boldsymbol{v}, \boldsymbol{z}\rangle - \langle\boldsymbol{v}, \boldsymbol{z}\rangle^2)].$$

We continue by showing that the expression inside the expectation is smaller than 1:

$$\begin{aligned}
\frac{1}{2}\exp(-y\langle\boldsymbol{v}, \boldsymbol{z}\rangle - \langle\boldsymbol{v}, \boldsymbol{z}\rangle^2) + \frac{1}{2}\exp(y\langle\boldsymbol{v}, \boldsymbol{z}\rangle - \langle\boldsymbol{v}, \boldsymbol{z}\rangle^2) &\leq 1 \\
\ln(\cosh(y\langle\boldsymbol{v}, \boldsymbol{z}\rangle))) - \langle\boldsymbol{v}, \boldsymbol{z}\rangle^2 &\leq 0.
\end{aligned}$$

which holds because for $|y| \leq 1.4$ $f(x) = \ln(\cosh(yx)) - x^2$ is concave and maximized at $x = 0$, which gives $f(0) = 0$. □

*Proof.* (of Lemma 5) Let $\tilde{\ell}_t(v) = v\tilde{g}_t + (v\tilde{g}_t)^2$

$$\mathbb{E}_{\tilde{g}_t}[F_t(-\sum_{s=1}^{t}\tilde{g}_s)] = \mathbb{E}_v[\mathbb{E}_{\tilde{g}_t}[\exp(-\tilde{\ell}_t(v) - \sum_{s=1}^{t-1}\tilde{\ell}_t(v)) - 1]]$$

$$\leq \mathbb{E}_v[(1 - v\,\mathbb{E}[\tilde{g}_t])\exp(-\sum_{s=1}^{t-1}\tilde{\ell}_t(v)) - 1]]$$

$$= F_{t-1}(-\sum_{s=1}^{t-1}\tilde{g}_s) - w_t\,\mathbb{E}[\tilde{g}_t]$$

where the first equality is due to Tonelli's theorem and the inequality is due to Lemma 4, which applies due to the restrictions on $v$ and $\mathbb{E}[\tilde{g}_t]$. Since $F_0(x) = 0$ the proof is complete. $\qquad\square$

### B.1 Regret Analysis for Proper Priors

*Proof.* (of Theorem 1). By Lemma 2, Lemma 3, and Lemma 5 we only have to compute the convex conjugate of the potential function. We do the analysis for $-\sum_{t=1}^{T}\tilde{g}_t \geq 0$. The analysis for $-\sum_{t=1}^{T}\tilde{g}_t \leq 0$ is analogous. We have $-\sum_{t=1}^{T}w_t\tilde{g}_t \geq F_T(-\sum_{t=1}^{T}\tilde{g}_t) \geq -1$. Suppose $\sum_{t=1}^{T}\tilde{g}_t \leq \sqrt{2(\sum_{t=1}^{T}\tilde{g}_t^2 + b)}$, then $\mathbb{E}[\mathcal{R}_T(u)] = \mathbb{E}[\sum_{t=1}^{T}w_t\tilde{g}_t - u\tilde{g}_t] \leq \mathbb{E}[\sum_{t=1}^{T}|u||\sum_{t=1}^{T}\tilde{g}_t|] + 1 \leq |u|\,\mathbb{E}[\sqrt{2(\sum_{t=1}^{T}\tilde{g}_t^2 + b)}] + 1$, which implies the result.

Now, suppose $\sum_{t=1}^{T}\tilde{g}_t \geq \sqrt{2(\sum_{t=1}^{T}\tilde{g}_t^2 + b)}$. For the conjugate prior $\nu([\eta, \mu]) = \eta - \mu$ and $Z \leq \frac{\sqrt{\pi}}{\sqrt{b}}$. In the case where $-\sum_{t=1}^{T}\tilde{g}_t \leq \frac{2}{5G}(\sum_{t=1}^{T}\tilde{g}_t^2 + b)$ set $\mu = \frac{-\sum_{t=1}^{T}\tilde{g}_t}{2(\sum_{t=1}^{T}\tilde{g}_t^2 + b)}$. Using Lemma 8 we obtain:

$$F_T^\star(u) \leq \sqrt{8|u|^2\left(\sum_{t=1}^{T}\tilde{g}_t^2 + b\right)\ln(16|u|^2\left(\sum_{t=1}^{T}\tilde{g}_t^2 + b\right)\sqrt{\pi}\frac{\sqrt{\sum_{t=1}^{T}\tilde{g}_t^2 + b}}{\sqrt{b}} + 1) + 1.} \quad (9)$$

In the case where $-\sum_{t=1}^{T}\tilde{g}_t \geq \frac{2}{5G}(\sum_{t=1}^{T}\tilde{g}_t^2 + b)$ set $\eta = \frac{5-\sqrt{5}}{50G}$ and $\mu = \frac{1}{2}$ to obtain:

$$F_T^\star(u) \leq 11G|u|(\ln(|u|11G) - 1 + \ln\left(\frac{\sqrt{5}G\sqrt{\pi}}{4\sqrt{b}}\right)) + 1. \quad (10)$$

Combining the expectations of (9) and (10) completes the proof. $\qquad\square$

**Lemma 8.** *Suppose $L > \sqrt{2(V+b)}$. Let $F_T(L) = \mathbb{E}_{v\sim P}[\exp(vL - v^2V) - 1]$ with $P$ as in (6). If $L \leq \frac{2}{5G}(V+b)$ then*

$$F_T^\star(u) \leq \sqrt{8|u|^2(V+b)\ln(16|u|^2(V+b)\tilde{\mathcal{R}}_t([\eta_1, \mu_1]) + 1)} + 1,$$

*where $\tilde{\mathcal{R}}_t([\eta, \mu]) = \frac{Z}{\nu([\eta,\mu])}$, $\eta_1 = \frac{L}{2(V+b)} - \frac{1}{\sqrt{2(V+b)}}$, $|\mu_1| \in [\eta_1, \frac{1}{5G}]$ such that $\mu_1 \leq \frac{L}{2(V+b)}$, and $\nu([\eta, \mu]) = \int_\eta^\mu \nu(v)dv$. If $L \geq \frac{2}{5G}(V+b)$ then*

$$F_T^\star(u) \leq \frac{|u|}{\eta - \eta^2\frac{5}{2}G}(\ln\left(\frac{|u|}{\eta_2 - \eta_2^2\frac{5}{2}G}\right) - 1 + \ln(\tilde{\mathcal{R}}_T([\eta_2, \mu_2]))) + 1,$$

*where $[\eta_2, \mu_2] \subseteq [-\frac{1}{5G}, \frac{1}{5G}]$ such that $\mu_2 \leq \frac{L}{2(V+b)}$.*

*Proof.* The initial part analysis is parallel to the analysis of Theorem 3 by Koolen and van Erven (2015). Denote by $B = V + b$. For $v \leq \hat{\eta} = \frac{L}{2B}$, $vL - v^2B$ is non-decreasing in $v$. Therefore, for

$[\eta, \mu] \subseteq [-\frac{1}{5G}, \frac{1}{5G}]$ such that $\mu \leq \hat{\eta}$:

$$F_T(-\sum_{t=1}^{T} x_t) = \frac{1}{Z} \int_{-\frac{1}{5G}}^{\frac{1}{5G}} \nu(v) \exp(vL - v^2 B) dv - 1$$

$$\geq \frac{1}{Z} \nu([\eta, \mu]) \exp(\eta L - \eta^2 B) - 1,$$

where $\nu([\eta, \mu]) = \int_{\eta}^{\mu} \nu(v) dv$. First suppose that $\hat{\eta} \leq \frac{1}{5G}$. Take $\eta = \hat{\eta} - \frac{1}{\sqrt{2B}}$, which yields

$$F_T(L) \geq \frac{\nu([\eta, \mu])}{Z} \exp\left(\frac{L^2}{4B} - \frac{1}{2}\right) - 1 = g(m(L)) - 1$$

where $g(x) = \exp(x - \frac{1}{2} - \ln\left(\frac{Z}{\nu([\eta,\mu])}\right))$ and $m(x) = \frac{x^2}{4B}$. By Hiriart-Urruty (2006, Theorem 2) we have

$$\begin{aligned}
F_T^\star(u) \leq (g(m(u)))^\star &= \inf_{\gamma \geq 0} g^\star(\gamma) + \gamma m^\star\left(\frac{u}{\gamma}\right) \\
&= \inf_{\gamma \geq 0} \gamma \ln(\gamma) + \gamma(\ln(\frac{Z}{\nu([\eta, \mu])}) - \frac{1}{2}) + \frac{1}{\gamma} 4|u|^2 B + 1.
\end{aligned} \quad (11)$$

Denote by $S = \ln(\frac{Z}{\nu([\eta,\mu])})$ and $H = 4|u|^2 B$. Setting the derivative to 0 we find that $\hat{\gamma} = \sqrt{\frac{2H}{W(2H \exp(\mathcal{R}_T^a + \frac{1}{2}))}}$ minimizes (11), where $W$ is the Lambert function. Plugging $\hat{\gamma}$ in (11) gives

$$F_T^\star(u) \leq \frac{H(2W(2H \exp(S + \frac{1}{2})) - 1)}{\sqrt{2H(W(2H \exp(S + \frac{1}{2})))}} + 1 \leq \sqrt{2H(W(2H \exp(S + \frac{1}{2})) + 1}.$$

Using $W(x) \leq \ln(x + 1)$ (Orabona and Pál, 2016, Lemma 17) we obtain

$$F_T^\star(u) \leq \sqrt{2H \ln(2H \exp(S + \frac{1}{2}) + 1)} \leq \sqrt{8|u|^2 B \ln(16|u|^2 B \exp(S) + 1)} + 1.$$

Now suppose that $\hat{\eta} > \frac{1}{5G}$, which is equivalent to $\frac{5}{2} GL > B$. Then

$$F_T(L) \geq \frac{\nu([\eta, \mu])}{Z} \exp((\eta - \eta^2 \frac{5}{2} G)L) - 1.$$

The convex conjugate of this lower bound is well known and is an upper bound on $F_T^\star$:

$$F_T^\star(u) \leq \frac{|u|}{\eta - \eta^2 \frac{5}{2} G} (\ln\left(\frac{|u|}{\eta - \eta^2 \frac{5}{2} G}\right) - 1 + \ln\left(\frac{Z}{\nu([\eta, \mu])}\right)) + 1,$$

which concludes the proof. $\qquad\square$

## B.2 Details From section 3.1

*Proof.* (of Lemma 6) We have

$$\begin{aligned}
\mathbb{E}[\mathcal{R}_u(\boldsymbol{u})] &= \mathbb{E}\left[\sum_{t=1}^{T} \langle \boldsymbol{w}_t - \boldsymbol{u}, \tilde{\boldsymbol{g}}_t \rangle\right] \\
&= \mathbb{E}\left[\sum_{t=1}^{T} \langle z_t, \tilde{\boldsymbol{g}}_t \rangle (v_t - \|\boldsymbol{u}\|)\right] + \|\boldsymbol{u}\| \mathbb{E}\left[\sum_{t=1}^{T} \langle z_t - \frac{\boldsymbol{u}}{\|\boldsymbol{u}\|}, \tilde{\boldsymbol{g}}_t \rangle\right] \\
&= \mathcal{R}_T^{\mathcal{V}}(\|\boldsymbol{u}\|) + \|\boldsymbol{u}\| \mathcal{R}_T^{\mathcal{Z}}\left(\frac{\boldsymbol{u}}{\|\boldsymbol{u}\|}\right)
\end{aligned}$$

$\qquad\square$

## C  Regret Analysis for the Improper Prior

Abbreviating $B_t = \sum_{s=1}^{t-1} \tilde{g}_s^2$, $L_t = -\sum_{s=1}^{t-1} \tilde{g}_s$, and $C = \frac{1}{5G}$, the predictions (5) with the improper prior are given by:

$$\frac{\sqrt{\pi} \exp(\frac{L^2}{4B}) \left(2 \operatorname{erf}\left(\frac{L}{2\sqrt{B}}\right) - \operatorname{erf}\left(\frac{L+2CB}{2\sqrt{B}}\right) - \operatorname{erf}\left(\frac{L-2CB}{2\sqrt{B}}\right)\right)}{2\sqrt{B}}. \tag{12}$$

With the predictions in (12) we can show the following result.

**Theorem 3.** *Suppose $\tilde{g}_t$ is a symmetrical random variable with $|\mathbb{E}[\tilde{g}_t]| \leq G$ for all t. The the expected regret of algorithm 1 with the improper prior $\frac{dP}{dv} = \frac{1}{|v|}$ satisfies*

$$\mathbb{E}[\mathcal{R}_T(u)] \leq \max \left\{ |u| \, \mathbb{E}\left[\sqrt{8 \sum_{t=1}^{T} \tilde{g}_t^2} \left(\sqrt{\ln(8|u|^2 \sum_{t=1}^{T} \tilde{g}_t^2 + 1)} + 1\right)\right], \right.$$
$$|u| 11 G (\ln(|u| 11 G \ln(2)) - 1) + \ln(2), \tag{13}$$
$$\left. |u| \, \mathbb{E}[\sqrt{2V}] + 1 + \mathbb{E}\left[\ln\left(1 + 2\sqrt{2V}\right)\right] \right\}.$$

*Proof.* By Lemma 2, Lemma 3, and Lemma 5 we only have to compute the convex conjugate of the potential function. The initial part analysis is parallel to the analysis Theorem 4 by Koolen and van Erven (2015). Denote by $L = -\sum_{t=1}^{T} \tilde{g}_t$ and by $V \sum_{t=1}^{T} \tilde{g}_t^2$. We do the analysis for $L \geq 0$. The analysis for $L \leq 0$ is analogous. We start by considering the case where $L \leq \sqrt{2V}$. We have

$$F_T(L) \geq \int_0^\epsilon \frac{1}{v}(\exp(-vL - v^2 V) - 1) + \int_\epsilon^{\frac{1}{5G}} \frac{1}{v}(\exp(-vL - v^2 V) - 1) \geq -\epsilon L - \epsilon^2 V + \ln(5G\epsilon),$$

where we used $\exp(x) \geq 1 + x$. Choosing $\epsilon = \frac{1}{5G + 2\sqrt{2V}}$ gives $-\mathbb{E}[\sum_{t=1}^{T} w_t \tilde{g}_t] \geq \mathbb{E}[F_T(L)] \geq -1 - \mathbb{E}[\ln\left(1 + 2\sqrt{2V}\right)]$. Now, $\mathbb{E}[\mathcal{R}_T(u)] = \mathbb{E}[\sum_{t=1}^{T} w_t \tilde{g}_t - u\tilde{g}_t] \leq \mathbb{E}[\sum_{t=1}^{T} |u||L|] + 1 + \mathbb{E}[\ln\left(1 + 2\sqrt{2V}\right)] \leq |u| \mathbb{E}[\sqrt{2V}] + 1 + \mathbb{E}[\ln\left(1 + 2\sqrt{2V}\right)]$.

Now consider the case where $L > \sqrt{2V}$. For $v \leq \hat{\eta} = \frac{L}{2V}$, $vL - v^2 V$ is non-decreasing in $v$. Therefore, for $[\eta, \mu] \subseteq [0, \frac{1}{5G}]$ such that $\mu \leq \hat{\eta}$, we have:

$$F_T(L) = \int_{-\frac{1}{5G}}^{\frac{1}{5G}} \frac{1}{|v|}(\exp(vL - v^2 V) - 1)dv$$
$$\geq (\exp(\eta L - \eta^2 V) - 1) \int_\eta^\mu \frac{1}{v} dv - \int_\mu^{\frac{1}{5G}} \frac{1}{v} dv$$
$$= (\exp(\eta L - \eta^2 V) - 1) \ln\left(\frac{\mu}{\eta}\right) + \ln(5G\mu).$$

First, suppose that $\hat{\eta} \leq \frac{1}{5G}$. Set $\mu = \hat{\eta}$ and $\eta = \hat{\eta} - \frac{1}{\sqrt{2V}}$ and use $L \geq 2\sqrt{V}$ to obtain

$$F_T(L) \geq \exp\left(\frac{L^2}{4V} - \frac{1}{2}\right) \ln\left(\frac{1}{1 - \frac{\sqrt{2V}}{L}}\right) + \ln\left(\frac{L}{V}\right)$$
$$\geq \exp\left(\frac{L^2}{4V} - \frac{1}{2}\right) \ln\left(\frac{1}{1 - \frac{\sqrt{2V}}{L}}\right) - \frac{1}{2}\ln\left(\frac{V}{4}\right)$$
$$\geq \exp\left(\frac{1}{2}\left(\frac{L}{\sqrt{2V}} - 1\right)^2\right) - 1,$$

where the last inequality follows by using $\exp\left(\frac{1}{2}(x^2-1)\right) \geq \exp\left(\frac{1}{2}(x-1)^2\right) x, -1 \geq -\frac{L}{\sqrt{2V}}$, and $-\ln(1-x) \geq x$. Write $\exp\left(\frac{1}{2}\left(\frac{L}{\sqrt{2V}}-1\right)^2\right) - 1 = g(m(x))$, where $g(x) = \exp(x) - 1$ and $m(x) = \left(\frac{x}{\sqrt{2V}}-1\right)^2$. By Hiriart-Urruty (2006, Theorem 2) we have

$$
\begin{aligned}
F_T^\star(u) \leq (g(m(u)))^\star &= \inf_{\gamma \geq 0} g^\star(\gamma) + \gamma m^\star(\frac{u}{\gamma}) \\
&= \inf_{\gamma \geq 0} \gamma \ln(\gamma) - \gamma + \frac{1}{\gamma} 4|u|^2 V + 2|u|\sqrt{2V}.
\end{aligned}
\tag{14}
$$

Setting the derivative to 0 we find that $\hat{\gamma} = \exp\left(\frac{1}{2}W(8|u|^2|V)\right)$ minimizes (14), where $W$ is the Lambert function. Plugging $\hat{\gamma}$ in (14) gives

$$
F_T^\star(u) \leq |u|\sqrt{8VW(8|u|^2|V)} - \hat{\gamma} + 2|u|\sqrt{2V}.
$$

Using $W(x) \leq \ln(x+1)$ (Orabona and Pál, 2016, Lemma 17) and dropping the negative term we obtain

$$
F_T^\star(u) \leq |u|\sqrt{8V}\left(\sqrt{\ln(8|u|^2V+1)}+1\right).
$$

Now suppose that $\hat{\eta} > \frac{1}{5G}$. Using that $\frac{5G}{2}L \geq V$, choosing $\mu = \frac{1}{5G}$, and $\eta = \frac{5-\sqrt{5}}{50G}$ we obtain

$$
\begin{aligned}
F_T(L) &\geq (\exp(\left(\frac{2(\sqrt{5}-1)}{25G}\right)L) - 1)\ln\left(\frac{1}{1-\frac{1}{\sqrt{5}}}\right) \\
&\geq (\exp(\left(\frac{1}{11G}\right)L) - 1)\ln(2).
\end{aligned}
\tag{15}
$$

The convex conjugate of the last expression in (15) is well known and given by

$$
F_T^\star(u) \leq |u|11G(\ln(|u|11G\ln(2)) - 1) + \ln(2).
$$

Combining the above completes the proof. $\qquad\square$