[Reviews · NeurIPS 2019]

Reviewer 1



Summary: The paper investigates an online convex optimization problem where gradients of loss functions are given combined with a random noise form an unknown symmetric distribution and the norm of the gradients are not bounded. The authors propose an algorithm which achieves state-of-the-art regret bounds in terms of the expected norm in the new noisy settings. The algorithm consists of a reduction to a one-dimensional OCO (coin betting) and an OCO. Comments: The theoretical analyses are solid and strong and subsume previous work, e.g., Jun-Orabona-COLT19. However, I feel that the topic and results of this paper looks quite similar to the previous COLT paper. In fact, algorithms are also similar. I also wonder how much the local differential private setting is related to the noisy OCO setting and I am afraid the connection is not so strong. Overall, the theoretical results are beyond standard ones, but the similarity to Jun-Orabona-COLT19 makes me feel that the contribution is a bit incremental. After reading the rebuttal comments, I understand the technical contribution more and raised my score.

Reviewer 2



Achieving data dependent bounds for OCO is an important topic in recent years, and it has been solved for OCO with bounded gradients. For noisy OCO, a recent COLT paper makes some progress towards the desired goal, which achieves a competitor dependent bound. However, that result still depends on Lipschitz constant rather than noisy gradients received in the game. In this paper, authors solved above problem for a special but important case of noisy OCO, which requires symmetric random gradients. The main framework is based on reward-regret duality paradigm with a carefully chosen potential function in terms of symmetric random gradients. Based on above adaptive noisy OCO algorithm, authors used it to solve OCO under LDP constraint, which naturally satisfies the condition about symmetric random gradients, and obtained corresponding data dependent bound. Besides, authors also designed a sparse gradient version, which might achieve better performance when the competitor or sub-gradients are sparse. Though how to achieve a completely data dependent regret bound for general noisy OCO is not clear yet, this paper solves the problem for a quite important case (i.e. symmetric random gradients), which can be directly used for OCO under LDP constraint and achieve corresponding data dependent bound. For this reason, I tend to accept this paper. Below are some comments and suggestions: 1. In equation (1), $\sigma_t$ should be $\sigma_t^2$; 2. In line 196-197, what does the word “multiplying 1-E[v\tilde{g}_t] for t=1,2,…” mean? Please make it clear; 3. How is the concrete update form of w_t (i.e. equation (7) in main part and equation (4) in appendix) obtained? Please make them clear; 4. In line 240, “turn in into” should be “turn it into”; 5. In line 260, “use” is redundant; 6. In line 290, I feel confused about the claim “\xi_t can be sparse as well”. Since \tau_{t,j} determines the degree of noise \xi_{t,j} and there is an upper bound of each \tau_{t,j} (say privacy budget \eps), \xi_{t,j} cannot be arbitrarily small, thus I think \xi_t is still a dense vector.

Reviewer 3



The paper deals with an interesting topic. Proposal is novel, sound, supported by theoretical results and convincing.

[Author Response · NeurIPS 2019]

Dear referees and chairs,

We thank all referees for their close reading of our manuscript.

Reviewer #1:

The question on how OCO and Local Differential Privacy (LDP) are related is an important one. In training many
machine learning models use an OCO algorithm. Without appropriate OCO algorithms these models can not be trained
with LDP guarantees. Since noisy OCO perfectly captures the requirements to satisfy LDP guarantees noisy OCO and
LDP seem to be a perfect fit. Because our bounds are adaptive to the unkown noise, data, and comparator our work is a
step towards practically useful algorithms with LDP guarantees that have sound theoretical guarantees. We will make
this connection more clear in the final version of the paper.

As mentioned by reviewer #3, designing data dependent bounds has been an important research topic in recent years.
Results in the noiseless setting have been transitioning from the traditional worst case optimal $O(\|\boldsymbol{u}\|\sqrt{T})$ results to
the more recent data dependent $O(\|\boldsymbol{u}\|\sqrt{\sum_{t=1}^{T}\|\boldsymbol{g}_t\|_\star^2})$. In the noisy setting we also have to adapt to the unknown
parameters of the distribution of the noise for data dependent bounds. The adaptivity to the unknown parameters of the
noise and data were open questions in the unconstrained setting before our paper.

Regarding novelty with respect to Jun and Orabona (2019): At a high level there are two similarities: 1) the use of the
reward-regret duality (section 2.3) and 2) the use of the black-box reduction (section 3.1). Indeed, these techniques
are cornerstones of much recent work in adaptive OCO. An early version of the reward-regret duality was introduced
by (Mcmahan and Streeter, 2012) and has been used in for example McMahan and Orabona (2014); Orabona and
Pál (2016); Orabona and Tommasi (2017); Cutkosky and Orabona (2018); and the black-box reduction comes from
Cutkosky and Orabona (2018) and was also used by Jun and Orabona (2019).

Even with these techniques in hand, what was not known before our work, is how to obtain results that allow for
different levels of differential privacy per user and per dimension and obtain data dependent bounds at the same time.
As we mention in lines 107/108 of the paper, a partial result can be achieved by extending the techniques of Jun and
Orabona (2019), but this result would be unsatisfactory, because their techniques crucially rely on knowing all the
differential privacy parameters of the noise. Furthermore, this extension would still not allow for data-dependent bounds.
As we argue in lines 24-26, these differential privacy parameters are themselves privacy sensitive (knowing how much
someone cares about privacy may reveal that they are a celebrity for example), so we do not want to assume that they
are known. We get around this issue by replacing the assumption of known privacy parameters by the alternative
assumption that the noise has an arbitrary symmetric distribution for which we do not need to know the parameters
or even the shape. With this new assumption we can handle all standard randomizers that are used for LDP, like for
instance the Laplace randomizer.

Reviewer #2:

2. The "multiplying "$1 - \mathbb{E}[v\tilde{g}_t]$ for $t = 1, 2, \ldots$" means that we multiply the bound in Lemma 4 to find the potential in
equation (4).

3. The concrete form of the updates comes from working out the expectation in equation (5) for the conjugate and
improper priors.

6. Here we allow the user to set $\tau_j = \infty$. While this does not give LDP guarantees for all attributes it does give LDP
guarantees for attributes with $\tau_j < \infty$. One can imagine a situation in which part of the data is already public, but part
of it is not. For example, a particular user might not care for privacy on social media posts but could be concerned
about browsing history. Therefore, the user will set $\tau_j = \infty$ for $j$ corresponding to social media posts, but set $\tau_j$ to be
small for $j$ corresponding to browsing history.

Thank you for pointing out the typos, we will fix them in the final version.

Reviewer #3:

We thank you for the positive review. We will try to address your comments in the final version of the paper.

[Meta-Review · NeurIPS 2019]

Dear authors: the reviewers carefully considered your paper and also discussed their evaluation post-rebuttal. There was broad agreement that the results are solid, the paper is clearly written, and that the contribution is significant. We are happy to accept such papers at NeurIPS 2019. But please do carefully incorporate the reviewers detailed comments when preparing your final camera-ready document.